# Robust Out-of-distribution Detection for Neural Networks

**Jiefeng Chen,** [1] **Yixuan Li,** [1] **Xi Wu,** [2] **Yingyu Liang,** [1] **Somesh Jha** [1]

[1] University of Wisconsin-Madison
[2] Google
{jiefeng; sharonli}@cs.wisc.edu, wu.andrew.xi@gmail.com, {yliang; jha}@cs.wisc.edu

## Abstract

Detecting out-of-distribution (OOD) inputs is critical for safely deploying deep learning models in the real world. Existing approaches for detecting OOD examples work well when evaluated on benign in-distribution and OOD samples. However, in this paper, we show that existing detection mechanisms can be extremely brittle when evaluating on in-distribution and OOD inputs with minimal adversarial perturbations which don't change their semantics. Formally, we extensively study the problem of *Robust Out-of-Distribution Detection* on common OOD detection approaches, and show that state-of-the-art OOD detectors can be easily fooled by adding small perturbations to the in-distribution and OOD inputs. To counteract these threats, we propose an effective algorithm called ALOE, which performs robust training by exposing the model to both adversarially crafted inlier and outlier examples. Our method can be flexibly combined with, and render existing methods robust. On common benchmark datasets, we show that ALOE substantially improves the robustness of state-of-the-art OOD detection, with 58.4% AUROC improvement on CIFAR-10 and 46.59% improvement on CIFAR-100.

## Introduction

Out-of-distribution (OOD) detection has become an indispensable part of building reliable open-world machine learning models (Bendale and Boult 2015). An OOD detector is used to determine whether an input is from the training data distribution (in-distribution examples), or from a different distribution (OOD examples). Previous OOD detection methods are usually evaluated on benign in-distribution and OOD inputs (Hsu et al. 2020; Huang and Li 2021; Lee et al. 2018; Liang, Li, and Srikant 2017; Liu et al. 2020). Recently, some works have shown the existence of adversarial OOD examples, which are generated by slightly perturbing the clean OOD inputs to make the OOD detectors fail to detect them as OOD examples, and have proposed some robust OOD detection methods to address the issue of adversarial OOD examples (Sehwag et al. 2019; Hein, Andriushchenko, and Bitterwolf 2019; Meinke and Hein 2019; Bitterwolf, Meinke, and Hein 2020; Chen et al. 2021).

In this paper, we also consider the problem of robust OOD detection. Different from previous works, we not only con-

sider adversarial OOD examples, but also consider adversarial in-distribution examples, which are generated by slightly perturbing the clean in-distribution inputs and cause the OOD detectors to falsely reject them. We argue that both adversarial in-distribution examples and adversarial OOD examples can cause severe consequences if the OOD detectors fail to detect them, as illustrated in Figure 1.

Formally, we study the problem of *robust out-of-distribution detection* and reveal the lack of robustness of common OOD detection methods. We show that existing OOD detection algorithms can be easily attacked to produce mistaken OOD prediction under small adversarial perturbations (Papernot et al. 2016; Goodfellow, Shlens, and Szegedy 2014; Biggio et al. 2013; Szegedy et al. 2013). Specifically, we construct *adversarial in-distribution examples* by adding small perturbations to the in-distribution inputs such that the OOD detectors will falsely reject them; whereas *adversarial OOD examples* are generated by adding small perturbations to the OOD inputs such that the OOD detectors will fail to reject them. Different from the common notion, the adversarial examples in our work are meant to fool the OOD detectors $G(x)$, rather than the original image classification model $f(x)$. It is also worth noting that the perturbation is sufficiently small so that the visual semantics as well as true distributional membership remain the same. Yet worryingly, state-of-the-art OOD detectors can fail to distinguish between adversarial in-distribution examples and adversarial OOD examples. Although there are some works trying to make OOD detection robust to adversarial OOD examples, scant attention has been paid to making the OOD detectors robust against both the adversarial in-distribution examples and adversarial OOD examples. To the best of our knowledge, we are the first to consider the issue of adversarial in-distribution examples.

To address the challenge , we propose an effective method, ALOE, that improves the robust OOD detection performance. Specifically, we perform robust training by exposing the model to two types of perturbed adversarial examples. For in-distribution training data, we create a perturbed example by searching in its $\epsilon$-ball that maximizes the negative log likelihood. In addition, we also utilize an auxiliary unlabeled dataset as in (Hendrycks, Mazeika, and Dietterich 2018), and create corresponding perturbed outlier example by searching in its $\epsilon$-ball that maximizes the KL-divergence between model output and a uniform distribution. The overall

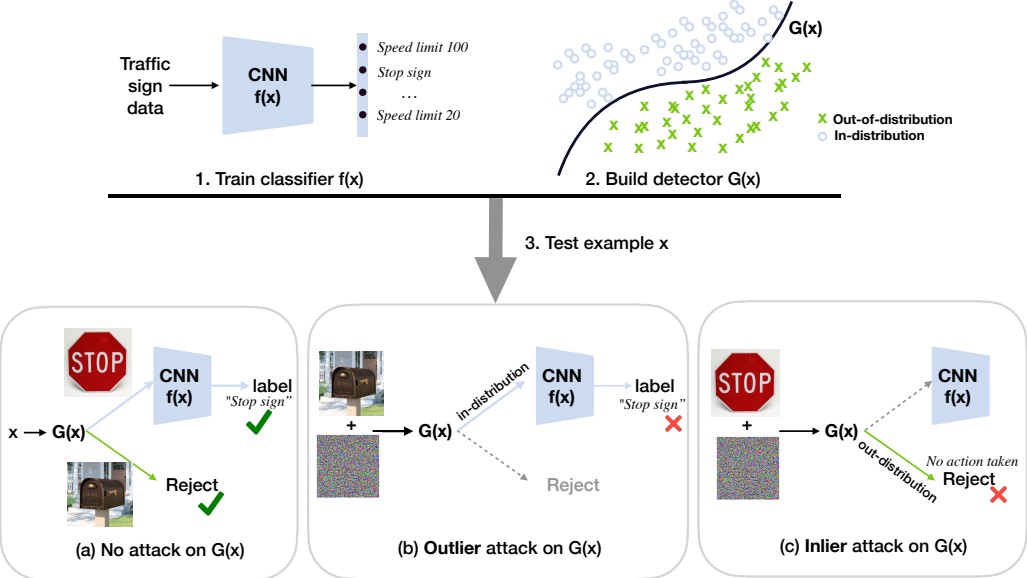

Figure 1: When deploying OOD detector $G(x)$ in the real world, there can be two types of attacks: outlier attack and inlier attack on $G(x)$. To perform outlier attack, we add small perturbation to an OOD input (e.g. mailbox) which causes the OOD detector to misclassify them as in-distribution example. The downstream classifier $f(x)$ will then classify this example into one of the known classes (e.g. stop sign), and trigger wrong action. To perform inlier attack, we add small perturbation to an in-distribution sample (e.g. stop sign) which causes the OOD detector to misclassify them as out-of-distribution example and reject it without taking the correct action (e.g. stop sign). Solid lines indicate the actual computation flow.

training objective of ALOE can be viewed as an adversarial min-max game. We show that on several benchmark datasets, ALOE can improve the robust OOD detection performance by up to 58.4% compared to previous state-of-the-art method. Our approach can be complemented by techniques such as ODIN (Liang, Li, and Srikant 2017), and further boost the performance.

Our main contributions are as follows:

- We extensively examine the robust OOD detection problem on common OOD detection approaches by considering both adversarial in-distribution examples and adversarial OOD examples. We show that state-of-the-art OOD detectors can fail to distinguish between in-distribution examples and OOD examples under small adversarial perturbations;
- We propose an effective algorithm, ALOE, that substantially improves the robustness of OOD detectors;
- We empirically analyze why common adversarial examples targeting the classifier with small perturbations should be regarded as in-distribution rather than OOD.
- We release a code base that integrates the most common OOD detection baselines, and our robust OOD detection methods at: https://github.com/jfc43/robust-ood-detection. We hope this can ensure reproducibility of all methods, and make it easy for the community to conduct future research on this topic.

## Related Work

**OOD Detection.** Hendrycks and Gimpel introduced a base-line for OOD detection using the maximum softmax probability from a pre-trained network. Subsequent works improve the OOD detection by using deep ensembles (Lakshminarayanan, Pritzel, and Blundell 2017), the calibrated softmax score (Liang, Li, and Srikant 2017), the Mahalanobis distance-based confidence score (Lee et al. 2018), and the energy score (Liu et al. 2020). Some methods also modify the neural networks by re-training or fine-tuning on some auxiliary anomalous data that are or realistic (Hendrycks, Mazeika, and Dietterich 2018; Mohseni et al. 2020) or artificially generated by GANs (Lee et al. 2017). Many other works (Subramanya, Srinivas, and Babu 2017; Malinin and Gales 2018; Bevandić et al. 2018) also regularize the model to have lower confidence on anomalous examples. Recent works have also studied the computational efficiency aspect of OOD detection (Lin, Roy, and Li 2021) and large-scale OOD detection on ImageNet (Huang and Li 2021).

**Robustness of OOD detection.** Worst-case aspects of OOD detection have previously been studied in (Sehwag et al. 2019; Hein, Andriushchenko, and Bitterwolf 2019; Meinke and Hein 2019; Bitterwolf, Meinke, and Hein 2020; Chen et al. 2021). However, these papers are primarily concerned with adversarial OOD examples. We are the first to present a unified framework to study both adversarial in-distribution examples and adversarial OOD examples.

**Adversarial Robustness.** A well-known phenomenon of adversarial examples (Biggio et al. 2013; Goodfellow, Shlens, and Szegedy 2014; Papernot et al. 2016; Szegedy et al. 2013) has received great attention in recent years. Many defense

methods have been proposed to address this problem. One of the most effective methods is adversarial training (Madry et al. 2017) which uses robust optimization techniques to render deep learning models resistant to adversarial attacks. In this paper, we show that the OOD detectors built from deep models are also very brittle under small perturbations, and propose a method to mitigate this issue using techniques from robust optimization.

## Traditional OOD Detection

Traditional OOD detection can be formulated as a canonical binary classification problem. Suppose we have an **in-distribution** $P_{\boldsymbol{X}}$ defined on an input space $\mathcal{X} \subset \mathbb{R}^n$. An OOD classifier $G : \mathcal{X} \mapsto \{0, 1\}$ is built to distinguish whether an input $x$ is from $P_{\boldsymbol{X}}$ (give it label 1) or not (give it label 0).

In testing, the detector $G$ is evaluated on inputs drawn from a mixture distribution $\mathcal{M}_{\boldsymbol{X} \times Z}$ defined on $\mathcal{X} \times \{0, 1\}$, where the conditional probability distributions $\mathcal{M}_{\boldsymbol{X}|Z=1} = P_{\boldsymbol{X}}$ and $\mathcal{M}_{\boldsymbol{X}|Z=0} = Q_{\boldsymbol{X}}$. We assume that $Z$ is drawn uniformly from $\{0, 1\}$. $Q_{\boldsymbol{X}}$ is also a distribution defined on $\mathcal{X}$ which we refer to it as **out-distribution**. Following previous work (Bendale and Boult 2016; Sehwag et al. 2019), we assume that $P_{\boldsymbol{X}}$ and $Q_{\boldsymbol{X}}$ are sufficiently different and $Q_{\boldsymbol{X}}$ has a label set that is disjoint from that of $P_{\boldsymbol{X}}$. We denote by $\mathcal{D}_{\text{in}}^{\text{test}}$ an in-distribution test set drawn from $P_{\boldsymbol{X}}$, and $\mathcal{D}_{\text{out}}^{\text{test}}$ an out-of-distribution test set drawn from $Q_{\boldsymbol{X}}$. The *detection error* of $G(x)$ evaluated under in-distribution $P_{\boldsymbol{X}}$ and out-distribution $Q_{\boldsymbol{X}}$ is defined by

$$L(P_{\boldsymbol{X}}, Q_{\boldsymbol{X}}; G) = \frac{1}{2}(\mathbb{E}_{x \sim P_{\boldsymbol{X}}}\mathbb{I}[G(x) = 0] \qquad (1)$$
$$+ \mathbb{E}_{x \sim Q_{\boldsymbol{X}}}\mathbb{I}[G(x) = 1])$$

## Robust Out-of-Distribution Detection

Traditional OOD detection methods are shown to work well when evaluated on natural in-distribution and OOD samples. However, in this section, we show that existing OOD detectors are extremely brittle and can fail when we add minimal semantic-preserving perturbations to the inputs. We start by formally describing the problem of *robust out-of-distribution detection*.

**Problem Statement.** We define $\Omega(x)$ to be a set of semantic-preserving perturbations on an input $x$. For $\delta \in \Omega(x)$, $x + \delta$ has the same semantic label as $x$. This also means that $x$ and $x + \delta$ have the same distributional membership (i.e. $x$ and $x + \delta$ both belong to in-distribution $P_{\boldsymbol{X}}$, or out-distribution $Q_{\boldsymbol{X}}$).

A robust OOD classifier $G : \mathcal{X} \mapsto \{0, 1\}$ is built to distinguish whether a perturbed input $x + \delta$ is from $P_{\boldsymbol{X}}$ or not. In testing, the detector $G$ is evaluated on perturbed inputs drawn from a mixture distribution $\mathcal{M}_{\boldsymbol{X} \times Z}$ defined on $\mathcal{X} \times \{0, 1\}$, where the conditional probability distributions $\mathcal{M}_{\boldsymbol{X}|Z=1} = P_{\boldsymbol{X}}$ and $\mathcal{M}_{\boldsymbol{X}|Z=0} = Q_{\boldsymbol{X}}$. We assume that $Z$ is drawn uniformly from $\{0, 1\}$. The *detection error* of $G$ evaluated under in-distribution $P_{\boldsymbol{X}}$ and out-distribution $Q_{\boldsymbol{X}}$ is now defined by

$$L(P_{\boldsymbol{X}}, Q_{\boldsymbol{X}}; G, \Omega) = \frac{1}{2}(\mathbb{E}_{x \sim P_{\boldsymbol{X}}} \max_{\delta \in \Omega(x)} \mathbb{I}[G(x + \delta) = 0]$$
$$+ \mathbb{E}_{x \sim Q_{\boldsymbol{X}}} \max_{\delta \in \Omega(x)} \mathbb{I}[G(x + \delta) = 1]) \quad (2)$$

In practice, it can be intractable to directly minimize $L(P_{\boldsymbol{X}}, Q_{\boldsymbol{X}}; G, \Omega)$ due to lack of prior knowledge on $Q_{\boldsymbol{X}}$. In some cases we assume having access to auxiliary data sampled from a distribution $U_{\boldsymbol{X}}$ which is different from both $P_{\boldsymbol{X}}$ and $Q_{\boldsymbol{X}}$.

**Adversarial Attacks on OOD Detection.** In the appendix, we describe a few common OOD detection methods such as MSP (Hendrycks and Gimpel 2016), ODIN (Liang, Li, and Srikant 2017) and Mahalanobis (Lee et al. 2018). We then propose adversarial attack algorithms that can show the vulnerability of these OOD detection approaches. Computing the exact value of detection error defined in equation (2) requires enumerating all possible perturbations. This can be practically intractable given the large space of $\Omega(x) \subset \mathbb{R}^n$. To this end, we propose adversarial attack algorithms that can find the perturbations in $\Omega(x)$ to compute a lower bound.

Specifically, we consider image data and small $L_\infty$ norm-bounded perturbations on $x$ since it is commonly used in adversarial machine learning research (Madry et al. 2017; Athalye, Carlini, and Wagner 2018). For data point $x \in \mathbb{R}^n$, a set of adversarial perturbations is defined as

$$B(x, \epsilon) = \{\delta \in \mathbb{R}^n \mid \|\delta\|_\infty \leq \epsilon \wedge x + \delta \text{ is valid}\}, \quad (3)$$

where $\epsilon$ is the size of small perturbation, which is also called adversarial budget. $x + \delta$ is considered valid if the values of $x + \delta$ are in the image pixel value range.

For the OOD detection methods based on softmax confidence score (e.g. MSP, ODIN and OE (Hendrycks, Mazeika, and Dietterich 2018)), we describe the attack mechanism in Algorithm 1. Specifically, we construct adversarial test examples by adding small perturbations in $B(x, \epsilon)$ so to change the prediction confidence in the reverse direction. To generate *adversarial in-distribution examples*, the model is induced to output probability distribution that is close to uniform; whereas *adversarial OOD examples* are constructed to induce the model produce high confidence score. We note here that the adversarial examples here are constructed to fool the OOD detectors $G(x)$, rather than the image classification model $f(x)$.

For the OOD detection methods using Mahalanobis distance based confidence score, we propose an attack algorithm detailed in Algorithm 2. Specifically, we construct adversarial test examples by adding small perturbations in $B(x, \epsilon)$ to make the logistic regression detector predict wrongly. Note that in our attack algorithm, we don't perform input pre-processing to compute the Mahalanobis distance based confidence score.

Our attack algorithms assume having access to the model parameters, thus they are white-box attacks. We find that using our attack algorithms, even with very minimal attack strength ($\epsilon = 1/255$ and $m = 10$), classic OOD detection methods (e.g. MSP, ODIN, Mahalanobis, OE, and

**Algorithm 1:** Adversarial attack on OOD detectors based on softmax confidence score.

**input** $x, F, \epsilon, m, \xi$
**output** $\delta$
    $\delta \leftarrow$ randomly choose a vector from $B(x, \epsilon)$
    **for** $t = 1, 2, \cdots, m$ **do**
        $x' \leftarrow x + \delta$
        **if** $x$ is in-distribution **then**
            $\ell(x') \leftarrow L_{\mathrm{CE}}(F(x'), \mathcal{U}_K)$
        **else**
            $\ell(x') \leftarrow -\sum_{i=1}^{K} F_i(x') \log F_i(x')$
        **end if**
        $\delta' \leftarrow \delta - \xi \cdot \mathrm{sign}(\nabla_x \ell(x'))$
        $\delta \leftarrow \prod_{B(x,\epsilon)} \delta'$         ▷ projecting $\delta'$ to $B(x, \epsilon)$
    **end for**

---

**Algorithm 2:** Adversarial attack on OOD detector using Mahalanobis distance based confidence score.

**input** $x, M_\ell(\cdot), \{\alpha_\ell\}, b, \epsilon, m, \xi$
**output** $\delta$
    $\delta \leftarrow$ randomly choose a vector from $B(x, \epsilon)$
    **for** $t = 1, 2, \cdots, m$ **do**
        $x' \leftarrow x + \delta$
        $p(x') \leftarrow \frac{1}{1 + e^{-(\sum_\ell \alpha_\ell M_\ell(x') + b)}}$
        **if** $x$ is in-distribution **then**
            $\ell(x') \leftarrow -\log p(x')$
        **else**
            $\ell(x') \leftarrow -\log(1 - p(x'))$
        **end if**
        $\delta' \leftarrow \delta + \xi \cdot \mathrm{sign}(\nabla_x \ell(x'))$
        $\delta \leftarrow \prod_{B(x,\epsilon)} \delta'$         ▷ projecting $\delta'$ to $B(x, \epsilon)$
    **end for**

---

OE+ODIN) can fail miserably. For example, the false positive rate of OE method can increase by 95.52% under such attack when evaluated on CIFAR-10 as in-distribution dataset.

## ALOE: Adversarial Learning with inliner and Outlier Exposure

In this section, we introduce a novel method called *Adversarial Learning with inliner and Outlier Exposure (ALOE)* to improve the robustness of the OOD detector $G(\cdot)$ built on top of the neural network $f(\cdot)$ against input perturbations.

**Training Objective.** We train our model ALOE against two types of perturbed examples. For in-distribution inputs $x \in P_{\boldsymbol{X}}$, ALOE creates *adversarial inlier* within the $\epsilon$-ball that maximize the negative log likelihood. Training with perturbed examples from the in-distribution helps calibrate the error on inliers, and make the model more invariant to the additive noise. In addition, our method leverages an auxiliary unlabeled dataset $\mathcal{D}_{\mathrm{out}}^{\mathrm{OE}}$ drawn from $U_{\boldsymbol{X}}$ as used in (Hendrycks, Mazeika, and Dietterich 2018), but in a different objective. While OE directly uses the original images $x \in \mathcal{D}_{\mathrm{out}}^{\mathrm{OE}}$ as outliers, ALOE creates *adversarial outliers* by searching within the $\epsilon$-ball that maximize the KL-divergence

between model output and a uniform distribution. The overall training objective of $F_{\mathrm{ALOE}}$ can be formulated as a min-max game given by

$$
\begin{aligned}
\underset{\theta}{\text{minimize}} \quad & \mathbb{E}_{(x,y) \sim \mathcal{D}_{\mathrm{in}}^{\mathrm{train}}} \max_{\delta \in B(x,\epsilon)} [-\log F_\theta(x+\delta)_y] \\
& + \lambda \cdot \mathbb{E}_{x \sim \mathcal{D}_{\mathrm{out}}^{\mathrm{OE}}} \max_{\delta \in B(x,\epsilon)} [L_{\mathrm{CE}}(F_\theta(x+\delta), \mathcal{U}_K)] \quad (4)
\end{aligned}
$$

where $F_\theta(x)$ is the softmax output of the neural network.

To solve the inner max of these objectives, we use the Projected Gradient Descent (PGD) method (Madry et al. 2017), which is the standard method for large-scale constrained optimization. The hyper-parameters of PGD used in the training will be provided in the experiments.

Once the model $F_{\mathrm{ALOE}}$ is trained, it can be used for downstream OOD detection by combining with approaches such as MSP and ODIN. The corresponding detectors can be constructed as $G_{\mathrm{MSP}}(x; \gamma, F_{\mathrm{ALOE}})$, and $G_{\mathrm{ODIN}}(x; T, \eta, \gamma, F_{\mathrm{ALOE}})$, respectively.

**Possible Variants.** We also derive two other variants of robust training objective for OOD detection. The first one performs adversarial training *only* on the inliers. We denote this method as ADV, which is equivalent to the objective used in (Madry et al. 2017). The training objective for ADV is:

$$
\underset{\theta}{\text{minimize}} \quad \mathbb{E}_{(x,y) \sim \mathcal{D}_{\mathrm{in}}^{\mathrm{train}}} \max_{\delta \in B(x,\epsilon)} [-\log F_\theta(x+\delta)_y]
$$

Alternatively, we also considered performing adversarial training on inlier examples while simultaneously performing outlier exposure as in (Hendrycks, Mazeika, and Dietterich 2018). We refer to this variant as AOE (adversarial learning with outlier exposure). The training objective for AOE is:

$$
\begin{aligned}
\underset{\theta}{\text{minimize}} \quad & \mathbb{E}_{(x,y) \sim \mathcal{D}_{\mathrm{in}}^{\mathrm{train}}} \max_{\delta \in B(x,\epsilon)} [-\log F_\theta(x+\delta)_y] \\
& + \lambda \cdot \mathbb{E}_{x \sim \mathcal{D}_{\mathrm{out}}^{\mathrm{OE}}} [L_{\mathrm{CE}}(F_\theta(x), \mathcal{U}_K)]
\end{aligned}
$$

We provide ablation studies comparing these variants with ALOE in the next section.

## Experiments

In this section we perform extensive experiments to evaluate previous OOD detection methods and our ALOE method under adversarial attacks on in-distribution and OOD inputs. Our main findings are summarized as follows:

**(1)** Classic OOD detection methods such as ODIN, Mahalanobis, and OE fail drastically under our adversarial attacks even with a very small perturbation budget.

**(2)** Our method ALOE can significantly improve the performance of OOD detection under our adversarial attacks compared to the classic OOD detection methods. Also, we observe that the performance of its variants ADV and AOE is worse than it in this task. And if we combine ALOE with other OOD detection approaches such as ODIN, we can further improve its performance. What's more, ALOE improves model robustness while maintaining almost the same classification accuracy on the clean test inputs (the results are in the appendix).

**(3)** Common adversarial examples targeting the image classifier $f(x)$ with small perturbations should be regarded as in-distribution rather than OOD.

Next we provide more details.

## Setup

**In-distribution Datasets.** we use GTSRB (Stallkamp et al. 2012), CIFAR-10 and CIFAR-100 datasets (Krizhevsky, Hinton et al. 2009) as in-distribution datasets. The pixel values of all the images are normalized to be in the range [0,1].

**Out-of-distribution Datasets.** For auxiliary outlier dataset, we use 80 Million Tiny Images (Torralba, Fergus, and Freeman 2008), which is a large-scale, diverse dataset scraped from the web. We follow the same deduplication procedure as in (Hendrycks, Mazeika, and Dietterich 2018) and remove all examples in this dataset that appear in CIFAR-10 and CIFAR-100 to ensure that $\mathcal{D}_{out}^{OE}$ and $\mathcal{D}_{out}^{test}$ are disjoint. For OOD test dataset, we follow the settings in (Liang, Li, and Srikant 2017; Hendrycks, Mazeika, and Dietterich 2018). For CIFAR-10 and CIFAR-100, we use six different natural image datasets: SVHN, Textures, Places365, LSUN (crop), LSUN (resize), and iSUN. For GTSRB, we use the following six datasets that are sufficiently different from it: CIFAR-10, Textures, Places365, LSUN (crop), LSUN (resize), and iSUN. Again, the pixel values of all the images are normalized to be in the range [0,1]. The details of these datasets can be found in the appendix.

**Architectures and Training Configurations.** We use the state-of-the-art neural network architecture DenseNet (Huang et al. 2017). We follow the same setup as in (Huang et al. 2017), with depth $L = 100$, growth rate $k = 12$ (Dense-BC) and dropout rate 0. All neural networks are trained with stochastic gradient descent with Nesterov momentum (Duchi, Hazan, and Singer 2011; Kingma and Ba 2014). Specifically, we train Dense-BC with momentum 0.9 and $\ell_2$ weight decay with a coefficient of $10^{-4}$. For GTSRB, we train it for 10 epochs; for CIFAR-10 and CIFAR-100, we train it for 100 epochs. For in-distribution dataset, we use batch size 64; For outlier exposure with $\mathcal{D}_{out}^{OE}$, we use batch size 128. The initial learning rate of 0.1 decays following a cosine learning rate schedule (Loshchilov and Hutter 2016).

**Hyperparameters.** For ODIN (Liang, Li, and Srikant 2017), we choose temperature scaling parameter $T$ and perturbation magnitude $\eta$ by validating on a random noise data, which does not depend on prior knowledge of out-of-distribution datasets in test. In all of our experiments, we set $T = 1000$. We set $\eta = 0.0004$ for GTSRB, $\eta = 0.0014$ for CIFAR-10, and $\eta = 0.0028$ for CIFAR-100. For Mahalanobis (Lee et al. 2018), we randomly select 1,000 examples from $\mathcal{D}_{in}^{train}$ and 1,000 examples from $\mathcal{D}_{out}^{OE}$ to train the Logistic Regression model and tune $\eta$, where $\eta$ is chosen from 21 evenly spaced numbers starting from 0 and ending at 0.004, and the optimal parameters are chosen to minimize the FPR at TPR 95%. For OE, AOE and ALOE methods, we fix the regularization parameter $\lambda$ to be 0.5. In PGD that solves the inner max of ADV, AOE and ALOE, we use step size $1/255$,

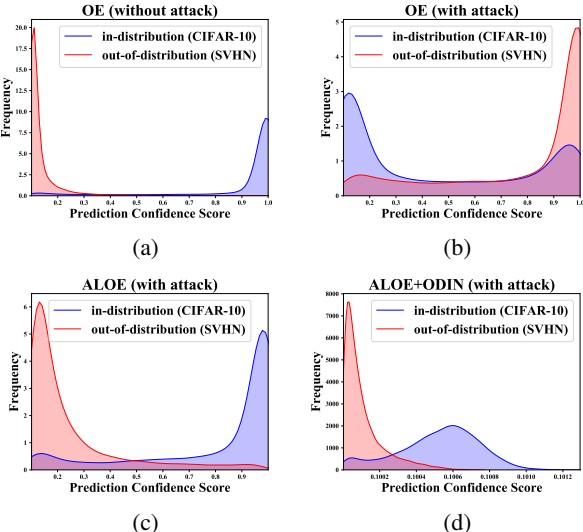

(a)         (b)

(c)         (d)

Figure 2: Confidence score distribution produced by different methods. For illustration purposes, we use CIFAR-10 as in-distribution and SVHN as out-of-distribution. (a) and (b) compare the score distribution for Outlier Exposure (Hendrycks, Mazeika, and Dietterich 2018), evaluated on clean images and PGD attacked images, respectively. The distribution overall shift toward the opposite direction under our attack, which causes the method to fail. Our method ALOE can mitigate the distribution shift as shown in (c). When combined with ODIN (Liang, Li, and Srikant 2017), the score distributions can be further separable between in- and out-distributions, as shown in (d).

number of steps $\lfloor 255\epsilon + 1 \rceil$, and random start. For our attack algorithm, we set $\xi = 1/255$ and $m = 10$ in our experiments. The adversarial budget $\epsilon$ by default is set to $1/255$, however we perform ablation studies by varying the value (see the results in the appendix).

More experiment settings can be found in the appendix.

## Evaluation Metrics

We report main results using three metrics described below.

**FPR at 95% TPR.** This metric calculates the false positive rate (FPR) on out-of-distribution examples when the true positive rate (TPR) is 95%.

**Detection Error.** This metric corresponds to the minimum mis-detection probability over all possible thresholds $\gamma$, which is $\min_\gamma L(P_X, Q_X; G(x; \gamma))$.

**AUROC.** Area Under the Receiver Operating Characteristic curve is a threshold-independent metric (Davis and Goadrich 2006). It can be interpreted as the probability that a positive example is assigned a higher detection score than a negative example (Fawcett 2006). A perfect detector corresponds to an AUROC score of 100%.

## Results

All the values reported in this section are averaged over *six* OOD test datasets.

Table 1 columns: $\mathcal{D}_{in}^{test}$, Method, then under "without attack": FPR (95% TPR)↓, Detection Error↓, AUROC↑, and under "with attack ($\epsilon = 1/255$, $m = 10$)": FPR (95% TPR)↓, Detection Error↓, AUROC↑.

| $\mathcal{D}_{in}^{test}$ | Method | FPR (95% TPR) ↓ | Detection Error ↓ | AUROC ↑ | FPR (95% TPR) ↓ | Detection Error ↓ | AUROC ↑ |
|---|---|---|---|---|---|---|---|
| | | without attack | | | with attack ($\epsilon = 1/255$, $m = 10$) | | |
| **GTSRB** | MSP (Hendrycks and Gimpel 2016) | 1.13 | 2.42 | 98.45 | 97.59 | 26.02 | 73.27 |
| | ODIN (Liang, Li, and Srikant 2017) | 1.42 | 2.10 | 98.81 | 75.94 | 24.87 | 75.41 |
| | Mahalanobis (Lee et al. 2018) | 1.31 | 2.87 | 98.29 | 100.00 | 29.80 | 70.45 |
| | OE (Hendrycks, Mazeika, and Dietterich 2018) | 0.02 | **0.34** | **99.92** | 25.85 | 5.90 | 96.09 |
| | OE+ODIN | 0.02 | 0.36 | 99.92 | 14.14 | 5.59 | 97.18 |
| | ADV (Madry et al. 2017) | 1.45 | 2.88 | 98.66 | 17.96 | 6.95 | 94.83 |
| | AOE | 0.00 | 0.62 | 99.86 | 1.49 | 2.55 | 98.35 |
| | ALOE (ours) | **0.00** | 0.44 | 99.76 | **0.66** | 1.80 | 98.95 |
| | ALOE+ODIN (ours) | 0.01 | 0.45 | 99.76 | 0.69 | **1.80** | **98.98** |
| **CIFAR-10** | MSP (Hendrycks and Gimpel 2016) | 51.67 | 14.06 | 91.61 | 99.98 | 50.00 | 10.34 |
| | ODIN (Liang, Li, and Srikant 2017) | 25.76 | 11.51 | 93.92 | 93.45 | 46.73 | 28.45 |
| | Mahalanobis (Lee et al. 2018) | 31.01 | 15.72 | 88.53 | 89.75 | 44.30 | 32.54 |
| | OE (Hendrycks, Mazeika, and Dietterich 2018) | 4.47 | 4.50 | 98.54 | 99.99 | 50.00 | 25.13 |
| | OE+ODIN | **4.17** | **4.31** | **98.55** | 99.02 | 47.84 | 34.29 |
| | ADV (Madry et al. 2017) | 66.99 | 19.22 | 87.23 | 98.44 | 31.72 | 66.73 |
| | AOE | 10.46 | 6.58 | 97.76 | 88.91 | 26.02 | 78.39 |
| | ALOE (ours) | 5.47 | 5.13 | 98.34 | 53.99 | 14.19 | 91.26 |
| | ALOE+ODIN (ours) | 4.48 | 4.66 | 98.55 | **41.59** | **12.73** | **92.69** |
| **CIFAR-100** | MSP (Hendrycks and Gimpel 2016) | 81.72 | 33.46 | 71.89 | 100.00 | 50.00 | 2.39 |
| | ODIN (Liang, Li, and Srikant 2017) | 58.84 | 22.94 | 83.63 | 98.87 | 49.87 | 21.02 |
| | Mahalanobis (Lee et al. 2018) | 53.75 | 27.63 | 70.85 | 95.79 | 47.53 | 17.92 |
| | OE (Hendrycks, Mazeika, and Dietterich 2018) | 56.49 | 19.38 | 87.73 | 100.00 | 50.00 | 2.94 |
| | OE+ODIN | **47.59** | **17.39** | **90.14** | 99.49 | 50.00 | 20.02 |
| | ADV (Madry et al. 2017) | 85.47 | 33.17 | 71.77 | 99.64 | 44.86 | 41.34 |
| | AOE | 60.00 | 23.03 | 84.57 | 95.79 | 43.07 | 53.80 |
| | ALOE (ours) | 61.99 | 23.56 | 83.72 | 92.01 | 40.09 | 61.20 |
| | ALOE+ODIN (ours) | 58.48 | 21.38 | 85.75 | **88.50** | **36.20** | **66.61** |

Table 1: Distinguishing in- and out-of-distribution test set data for image classification. We contrast performance on clean images (without attack) and PGD attacked images. ↑ indicates larger value is better, and ↓ indicates lower value is better. All values are percentages and are averaged over six OOD test datasets.

| $\mathcal{D}_{in}^{test}$ | Method | 1-FPR (95% TPR) |
|---|---|---|
| **CIFAR-10** | MSP (Hendrycks and Gimpel 2016) | 10.75 |
| | ODIN (Liang, Li, and Srikant 2017) | 4.02 |
| | Mahalanobis (Lee et al. 2018) | 7.13 |
| | OE (Hendrycks, Mazeika, and Dietterich 2018) | 12.22 |
| | OE+ODIN | 12.95 |
| | ADV (Madry et al. 2017) | 7.69 |
| | AOE | 11.18 |
| | ALOE (ours) | 8.85 |
| | ALOE+ODIN (ours) | 8.71 |
| **CIFAR-100** | MSP (Hendrycks and Gimpel 2016) | 0.06 |
| | ODIN (Liang, Li, and Srikant 2017) | 0.74 |
| | Mahalanobis (Lee et al. 2018) | 4.29 |
| | OE (Hendrycks, Mazeika, and Dietterich 2018) | 4.36 |
| | OE+ODIN | 5.21 |
| | ADV (Madry et al. 2017) | 3.14 |
| | AOE | 8.08 |
| | ALOE (ours) | 7.32 |
| | ALOE+ODIN (ours) | 7.06 |

Table 2: Distinguishing adversarial examples generated by PGD attack on the image classifier $f(x)$. 1-FPR indicates the rate of misclassifying adversarial examples as out-of-distribution examples. For PGD attack, we choose $\epsilon$ as $1/255$ and the number of attack steps as 10. All values are percentages.

**Classic OOD detection methods fail under our attack.**
As shown in Table 1, although classic OOD detection methods (e.g. MSP, ODIN, Mahalanobis, OE and OE+ODIN)

could perform quite well on detecting natural OOD samples, their performance drops substantially under the attack (even with very minimal attack budget $\epsilon = 1/255$ and $m = 10$). For the best-performing OOD detection method (i.e., OE+ODIN), the FPR at 95% TPR increases drastically from 4.17% (without attack) to 99.02% (with attack) when evaluated on the CIFAR-10 dataset.

**ALOE improves robust OOD detection performance.**
As shown in Table 1, our method ALOE could significantly improve the OOD detection performance under the adversarial attack. For example, ALOE can substantially improve the AUROC from 34.29% (state-of-the-art: OE+ODIN) to 92.69% evaluated on the CIFAR-10 dataset. The performance can be further improved when combining ALOE with ODIN. We observe this trend holds consistently on other benchmark datasets GTSRB and CIFAR-100 as in-distribution training data. We also find that adversarial training (ADV) or combining adversarial training with outlier exposure (AOE) yield slightly less competitive results.

To better understand our method, we analyze the distribution of confidence scores produced by the OOD detectors on SVHN (out-distribution) and CIFAR-10 (in-distribution). As shown in Figure 2, OE could distinguish in-distribution and out-of-distribution samples quite well since the confidence scores are well separated. However, under our attack, the confidence scores of in-distribution samples move towards 0

and the scores of out-of-distribution samples move towards 1.0, which renders the detector fail to distinguish in- and out-of-distribution samples. Using our method, the confidence scores (under attack) become separable and shift toward the right direction. If we further combine ALOE with ODIN, the scores produced by the detector are even more separated.

**Evaluating on common adversarial examples targeting the classifier** $f(x)$**.** Our work is primarily concerned with adversarial examples targeting OOD detectors $G(x)$. This is very different from the common notion of adversarial examples that are constructed to fool the image classifier $f(x)$. Based on our robust definition of OOD detection, adversarial examples constructed from in-distribution data with small perturbations to fool the image classifier $f(x)$ should be regarded as in-distribution. To validate this point, we generate PGD attacked images w.r.t the original classification model $f(x)$ trained on CIFAR-10 and CIFAR-100 respectively using a small perturbation budget of $1/255$. We measure the performance of OOD detectors $G(x)$ by reporting 1-FPR (at TPR 95%), which indicates the rate of misclassifying adversarial examples as out-of-distribution examples. As shown in Table 2, the metric in general is low for both classic and robust OOD detection methods, which suggests that common adversarial examples with small perturbations are closer to in-distribution rather than OOD.

## Conclusion

In this paper, we study the problem, Robust Out-of-Distribution Detection, and propose adversarial attack algorithms which reveal the lack of robustness of a wide range of OOD detection methods. We show that state-of-the-art OOD detection methods can fail catastrophically under both adversarial in-distribution and out-of-distribution attacks. To counteract these threats, we propose a new method called ALOE, which substantially improves the robustness of state-of-the-art OOD detection. We empirically analyze our method under different parameter settings and optimization objectives, and provide theoretical insights behind our approach. Future work involves exploring alternative semantic-preserving perturbations beyond adversarial attacks.

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

# Appendix

## Existing Approaches

Recently, several approaches propose to detect OOD examples based on different notions of confidence scores from a neural network $f(\cdot)$, which is trained on a dataset $\mathcal{D}_{\text{in}}^{\text{train}}$ drawn from a data distribution $P_{X,Y}$ defined on $\mathcal{X} \times \mathcal{Y}$ with $\mathcal{Y} = \{1, 2, \cdots, K\}$. Note that $P_X$ is the marginal distribution of $P_{X,Y}$. Based on this notion, we describe a few common methods below.

**Maximum Softmax Probability (MSP).** Maximum Softmax Probability method is as a common baseline for OOD detection (Hendrycks and Gimpel 2016). Given an input image $x$ and a pre-trained neural network $f(\cdot)$, the softmax output of the classifier is computed by $F(x) = \frac{e^{f_i(x)}}{\sum_{j=1}^{K} e^{f_j(x)}}$.

A threshold-based detector $G(x)$ relies on the confidence score $S(x; f) = \max_i F_i(x)$ to make prediction as follows

$$G_{\text{MSP}}(x; \gamma, f) = \begin{cases} 0 & \text{if } S(x; f) \leq \gamma \\ 1 & \text{if } S(x; f) > \gamma \end{cases} \quad (5)$$

where $\gamma$ is the confidence threshold.

**ODIN.** The original softmax confidence scores used in (Hendrycks and Gimpel 2016) can be over-confident. ODIN (Liang, Li, and Srikant 2017) leverages this insight and improves the MSP baseline using the calibrated confidence score instead (Guo et al. 2017). Specifically, the calibrated confidence score is computed by $S(x; T, f) = \max_i \frac{e^{f_i(x)/T}}{\sum_{j=1}^{K} e^{f_j(x)/T}}$, where $T \in \mathbb{R}^+$ is a temperature scaling

parameter. In addition, ODIN applies small noise perturbation to the inputs

$$\tilde{x} = x - \eta \cdot \text{sign}(-\nabla_x \log S(x; T, f)), \quad (6)$$

where the parameter $\eta$ is the perturbation magnitude.

By combining the two components together, ODIN detector $G_{\text{ODIN}}$ is given by

$$G_{\text{ODIN}}(x; T, \eta, \gamma, f) = \begin{cases} 0 & \text{if } S(\tilde{x}; T, f) \le \gamma \\ 1 & \text{if } S(\tilde{x}; T, f) > \gamma \end{cases} \quad (7)$$

In real applications, it may be difficult to know the out-of-distribution samples one will encounter in advance. The hyperparameters of $T$ and $\eta$ can be tuned instead on a random noise data such as Gaussian or uniform distribution, without requiring prior knowledge of OOD dataset.

**Mahalanobis.** Lee et al. model the features of training data as class-conditional Gaussian distribution, where its parameters are chosen as empirical class means and empirical covariance of training samples. Specifically, for a given sample $x$, the confidence score from the $\ell$-th feature layer is defined using the Mahalanobis distance with respect to the closest class-conditional distribution:

$$M_\ell(x) = \max_c -(f_\ell(x) - \hat{\mu}_{\ell,c})^T \hat{\Sigma}_\ell^{-1}(f_\ell(x) - \hat{\mu}_{\ell,c}), \quad (8)$$

where $f_\ell(x)$ is the $\ell$-th hidden features of DNNs, and $\hat{\mu}_{\ell,c}$ and $\hat{\Sigma}_\ell$ are the empirical class means and covariances computed from the training data respectively.

In addition, they use two techniques (1) input pre-processing and (2) feature ensemble. Specifically, for each test sample $x$, they first calculate the pre-processed sample $\tilde{x}_\ell$ by adding the small perturbations as in (Liang, Li, and Srikant 2017): $\tilde{x}_\ell = x + \eta \cdot \text{sign}(\nabla_x M_\ell(x))$, where $\eta$ is a magnitude of noise, which can be tuned on the validation data.

The confidence scores from all layers are integrated through a weighted averaging: $\sum_\ell \alpha_\ell M_\ell(\tilde{x}_\ell)$. The weight of each layer $\alpha_\ell$ is learned through a logistic regression model, which predicts 1 for in-distribution and 0 for OOD examples. The overall Mahalanobis distance based confidence score is

$$M(x) = \frac{1}{1 + e^{-(\sum_\ell \alpha_\ell M_\ell(\tilde{x}_\ell) + b)}}, \quad (9)$$

where $b$ is the bias of the logistic regression model. Putting it all together, the final Mahalanobis detector $G_{\text{Mahalanobis}}$ is given by

$$G_{\text{Mahalanobis}}(x; \eta, \gamma, \{\alpha_\ell\}, b, f) = \begin{cases} 0 & \text{if } M(x) \le \gamma \\ 1 & \text{if } M(x) > \gamma \end{cases}$$
$$(10)$$

## Experimental Details

### Setup

**Software and Hardware.** We run all experiments with PyTorch and NVDIA GeForce RTX 2080Ti GPUs.

**Number of Evaluation Runs.** We run all experiments once with fixed random seeds.

**In-distribution Dataset.** We provide the details of in-distribution datasets below:

1. **CIFAR-10 and CIFAR-100.** The CIFAR-10 and CIFAR-100 (Krizhevsky, Hinton et al. 2009) have 10 and 100 classes respectively. Both datasets consist of 50,000 training images and 10,000 test images.

2. **GTSRB.** The German Traffic Sign Recognition Benchmark (GTSRB) (Stallkamp et al. 2012) is a dataset of color images depicting 43 different traffic signs. The images are not of a fixed dimensions and have rich background and varying light conditions as would be expected of photographed images of traffic signs. There are about 34,799 training images, 4,410 validation images and 12,630 test images. We resize each image to $32 \times 32$. The dataset has a large imbalance in the number of sample occurrences across classes. We use data augmentation techniques to enlarge the training data and make the number of samples in each class balanced. We construct a class preserving data augmentation pipeline consisting of rotation, translation, and projection transforms and apply this pipeline to images in the training set until each class contained 10,000 training examples. This new augmented dataset containing 430,000 samples in total is used as $\mathcal{D}_{\text{in}}^{\text{train}}$. We randomly select 10,000 images from original test images as $\mathcal{D}_{\text{in}}^{\text{test}}$.

**OOD Test Dataset.** We provide the details of OOD test datasets below:

1. **SVHN.** The SVHN dataset (Netzer et al. 2011) contains $32 \times 32$ color images of house numbers. There are ten classes comprised of the digits 0-9. The original test set has 26,032 images. We randomly select 1,000 images for each class from the test set to form a new test dataset containing 10,000 images for our evaluation.

2. **Textures.** The Describable Textures Dataset (DTD) (Cimpoi et al. 2014) contains textural images in the wild. We include the entire collection of 5640 images in DTD and downsample each image to size $32 \times 32$.

3. **Places365.** The Places365 dataset (Zhou et al. 2017) contains large-scale photographs of scenes with 365 scene categories. There are 900 images per category in the test set. We randomly sample 10,000 images from the test set for evaluation and downsample each image to size $32 \times 32$.

4. **LSUN (crop) and LSUN (resize).** The Large-scale Scene UNderstanding dataset (LSUN) has a testing set of 10,000 images of 10 different scenes (Yu et al. 2015). We construct two datasets, LSUN-C and LSUN-R, by randomly cropping image patches of size $32 \times 32$ and downsampling each image to size $32 \times 32$, respectively.

5. **iSUN.** The iSUN (Xu et al. 2015) consists of a subset of SUN images. We include the entire collection of 8925 images in iSUN and downsample each image to size $32 \times 32$.

6. **CIFAR-10.** We use the 10,000 test images of CIFAR-10 as OOD test set for GTSRB.

| $\mathcal{D}_{in}^{test}$ | Method | FPR (95% TPR) ↓ | Detection Error ↓ | AUROC ↑ | FPR (95% TPR) ↓ | Detection Error ↓ | AUROC ↑ | FPR (95% TPR) ↓ | Detection Error ↓ | AUROC ↑ |
|---|---|---|---|---|---|---|---|---|---|---|
| | | with attack ($\epsilon = 2/255$, $m = 10$) | | | with attack ($\epsilon = 3/255$, $m = 10$) | | | with attack ($\epsilon = 4/255$, $m = 10$) | | |
| GTSRB | MSP (Hendrycks and Gimpel 2016) | 99.88 | 50.00 | 26.11 | 99.99 | 50.00 | 6.79 | 99.99 | 50.00 | 6.39 |
| | ODIN (Liang, Li, and Srikant 2017) | 99.23 | 49.97 | 27.38 | 99.83 | 50.00 | 6.94 | 99.84 | 50.00 | 6.52 |
| | Mahalanobis (Lee et al. 2018) | 100.00 | 49.97 | 26.37 | 100.00 | 50.00 | 8.27 | 100.00 | 50.00 | 7.82 |
| | OE (Hendrycks, Mazeika, and Dietterich 2018) | 96.79 | 16.09 | 83.06 | 99.91 | 25.36 | 68.62 | 99.97 | 26.37 | 66.91 |
| | OE+ODIN | 89.88 | 15.78 | 84.56 | 99.25 | 24.70 | 69.71 | 99.45 | 25.67 | 68.02 |
| | ADV (Madry et al. 2017) | 92.17 | 11.51 | 89.92 | 99.65 | 18.59 | 80.85 | 99.49 | 18.68 | 81.17 |
| | AOE | 7.94 | 5.36 | 94.82 | 16.16 | 10.38 | 88.72 | 38.05 | 17.95 | 83.84 |
| | ALOE (ours) | 4.03 | 4.19 | **95.90** | 10.82 | 7.64 | **91.21** | 16.10 | 10.10 | **89.52** |
| | ALOE+ODIN (ours) | **3.95** | **4.15** | 95.72 | **9.56** | **6.91** | 91.08 | **13.85** | **9.22** | 89.44 |
| CIFAR-10 | MSP (Hendrycks and Gimpel 2016) | 100.00 | 50.00 | 1.16 | 100.00 | 50.00 | 0.13 | 100.00 | 50.00 | 0.12 |
| | ODIN (Liang, Li, and Srikant 2017) | 99.73 | 49.99 | 5.67 | 99.98 | 50.00 | 1.14 | 99.99 | 50.00 | 1.06 |
| | Mahalanobis (Lee et al. 2018) | 100.00 | 50.00 | 5.90 | 100.00 | 50.00 | 1.27 | 100.00 | 50.00 | 1.05 |
| | OE (Hendrycks, Mazeika, and Dietterich 2018) | 100.00 | 50.00 | 5.99 | 100.00 | 50.00 | 1.52 | 100.00 | 50.00 | 1.48 |
| | OE+ODIN | 100.00 | 50.00 | 8.89 | 100.00 | 50.00 | 2.76 | 100.00 | 50.00 | 2.69 |
| | ADV (Madry et al. 2017) | 99.94 | 36.57 | 56.01 | 99.89 | 39.64 | 49.88 | 99.96 | 40.57 | 48.02 |
| | AOE | 91.79 | 35.08 | 66.92 | 99.96 | 39.53 | 54.43 | 98.40 | 37.37 | 59.16 |
| | ALOE (ours) | 75.90 | 23.36 | 83.26 | 83.14 | 31.54 | 73.46 | 82.53 | 29.92 | 75.52 |
| | ALOE+ODIN (ours) | **68.80** | **20.31** | **85.92** | **79.19** | **28.04** | **77.88** | **78.46** | **27.55** | **78.83** |

Table 3: Distinguishing in- and out-of-distribution test set data for image classification. ↑ indicates larger value is better, and ↓ indicates lower value is better. All values are percentages. The in-distribution datasets are GRSRB and CIFAR-10. All the values reported are averaged over six OOD test datasets.

| $\mathcal{D}_{in}^{test}$ | Method | Classifcation Accuracy | Robustness w.r.t image classifer |
|---|---|---|---|
| GTSRB | Original | 99.33% | 88.47% |
| | OE | 99.38% | 83.99% |
| | ADV | 99.23% | 97.13% |
| | AOE | 98.82% | 94.14% |
| | ALOE | 98.91% | 94.58% |
| CIFAR-10 | Original | 94.08% | 25.38% |
| | OE | 94.59% | 28.94% |
| | ADV | 92.97% | 84.81% |
| | AOE | 93.35% | 78.60% |
| | ALOE | 93.89% | 84.02% |
| CIFAR-100 | Original | 75.26% | 7.29% |
| | OE | 74.45% | 7.84% |
| | ADV | 70.58% | 54.58% |
| | AOE | 72.56% | 52.96% |
| | ALOE | 71.62% | 55.97% |

Table 4: The image classification accuracy and robustness of different models on original tasks (GTSRB, CIFAR-10 and CIFAR-100). *Robustness* measures the accuracy under PGD attack w.r.t the original classification model.

robustness on the original classification task. The results are presented in table 4. *Robustness* measures the accuracy under PGD attack w.r.t the original classification model. We use adversarial budget $\epsilon$ of $1/255$ and number of attack steps of 10. *Original* refers to the vanilla model trained with standard cross entropy loss on the dataset. On both GTSRB and CIFAR-10, ALOE improves the model robustness, while maintaining almost the same classification accuracy on the clean inputs. On CIFAR-100, ALOE improves robustness from 7.29% to 55.97%, albeit dropping the classification accuracy slightly (3.64%). Overall our method achieves good trade-off between the accuracy and robustness due to adversarial perturbations.

## Additional Results

**Effect of adversarial budget $\epsilon$.** We further perform ablation study on the adversarial budget $\epsilon$ and analyze how this affects performance. On GTSRB and CIFAR-10 dataset, we perform comparison by varying $\epsilon = 1/255, 2/255, 3/255, 4/255$. The results are reported in Table 3. We observe that as we increase $\epsilon$, the performance on classic OOD detection methods (e.g. MSP, ODIN, Mahalanobis, OE, OE+ODIN) drops significantly under our attack: the FPR at 95% TPR reaches almost 100% for all those methods. We also observe that our methods ALOE (and ALOE+ODIN) consistently improves the results under our attack compared to those classic methods.

**Classification performance of image classifier $f(x)$.** In addition to OOD detection, we also verify the accuracy and