# OpenReview forum: "Robust Out-of-distribution Detection for Neural Networks"
_AAAI.org/2022/Workshop/AdvML — AAAI-22 AdvML Workshop LongPaper_

### Official Review · Reviewer_NGPC · 2021-11-27

**Rating:** 7
**Confidence:** 4

**Review:**

**Summary of the paper:**

This paper extensively studies the problem of robust OOD detection on common OOD detection approaches and shows that existing OOD detection algorithms can be easily attacked to produce mistaken OOD prediction by adding small perturbations to the in-distribution and OOD inputs. To address the challenge, the authors propose an effective method to improve the robust OOD detection performance. Moreover, the authors show that the proposed method can improve the robust OOD detection performance by up to 58.4% compared to the previous state-of-the-art method on several benchmark datasets.

**Detailed comments:**

a.) This paper studies the problem of _Robust Out-of-Distribution Detection_ and shows that state-of-the-art OOD detectors can be easily fooled by adding small perturbations to the in-distribution and OOD inputs. The related analysis and conclusion are interesting and valuable if there is no similar analysis in the past.

b.) The proposed method performs robust training by exposing the model to both adversarially crafted inlier and outlier examples, which is simple yet effective and works well for a wide range of datasets.

---

### Official Review · Reviewer_ZnPo · 2021-11-30
**Lacks Novelty**

**Rating:** 5
**Confidence:** 3

**Review:**

**Summary**:

The paper talks about how adversarial perturbations can affect the performance of OOD detectors. They present a unified framework to study both adversarial attacks on in-distribution as well as OOD inputs (ie attacks which aim to increase both FP and FN). An algorithm called ALOE is presented that trains the model with perturbed inputs and improves robustness towards such attacks.


**PROS**

1. OOD detection is an important problem due to adversarial attacks and data drift in the real world, hence the problem is pertinent.
2. Evaluation is detailed and compares against well-known methods.


**CONS**

1. To me, the paper lacks innovation and novelty. Reduced to the basic version, the proposed approach is simply adversarial training with some minor tweaks. Exposing the model to perturbed inputs to improve robustness has been done previously, see [1] and [2] for examples. The only difference is the domain; while previous experiments were on classifiers, these are on OOD detectors (which are also fundamentally, classifiers).
2. The authors claim that one of the contributions is that they 'show that state-of-the-art OOD detectors can fail to distinguish between in-distribution examples and OOD examples under small adversarial perturbations'; however, this is not surprising, see [3].
3. Finally, the attacks presented are _white box_ which seems improbable. OOD detectors are generally not exposed as an endpoint, and most users do not even perceive that there is an OOD detector at work here. Therefore, I believe that white box attacks against OOD detectors are not of practical signifiance.


**References**

[1] Goodfellow, Ian J., Jonathon Shlens, and Christian Szegedy. "Explaining and harnessing adversarial examples." arXiv preprint arXiv:1412.6572 (2014).

[2] Wang, Y., Ma, X., Bailey, J., Yi, J., Zhou, B., & Gu, Q. (2019, June). On the Convergence and Robustness of Adversarial Training. In ICML (Vol. 1, p. 2).

[3] Sehwag, V., Bhagoji, A. N., Song, L., Sitawarin, C., Cullina, D., Chiang, M., & Mittal, P. (2019, November). Analyzing the robustness of open-world machine learning. In Proceedings of the 12th ACM Workshop on Artificial Intelligence and Security (pp. 105-116).

---

### Decision · Program_Chairs · 2021-12-01

**Decision:**

Accept (Long Paper)

**Comment:**

Reviewer ZnPo raises some concerns about the novelty and contributions of this paper. Based on the comments of both reviewers, this paper is accepted as a long paper. Please clarify the novelty and contributions in the final version.